# MicroServices Suite for Smart City Applications

**DOI:** 10.3390/s19214798

**Published:** 2019-11-04

**Authors:** Claudio Badii, Pierfrancesco Bellini, Angelo Difino, Paolo Nesi, Gianni Pantaleo, Michela Paolucci

**Affiliations:** UNIFI, DISIT Lab (Distributed Systems and Internet Technologies Lab), University of Florence, 50121 Florence, Italy; Claudio.badii@unifi.it (C.B.); pierfrancesco.bellini@unifi.it (P.B.); angelo.difino@unifi.it (A.D.); Gianni.pantaleo@unifi.it (G.P.); michela.paolucci@unifi.it (M.P.)

**Keywords:** smart city, IoT applications, MicroServices, node-RED, dashboard

## Abstract

Smart Cities are approaching the Internet of Things (IoT) World. Most of the first-generation Smart City solutions are based on Extract Transform Load (ETL); processes and languages that mainly support pull protocols for data gathering. IoT solutions are moving forward to event-driven processes using push protocols. Thus, the concept of IoT applications has turned out to be widespread; but it was initially “implemented” with ETL; rule-based solutions; and finally; with true data flows. In this paper, these aspects are reviewed, highlighting the requirements for smart city IoT applications and in particular, the ones that implement a set of specific MicroServices for IoT Applications in Smart City contexts. Moreover; our experience has allowed us to implement a suite of MicroServices for Node-RED; which has allowed for the creation of a wide range of new IoT applications for smart cities that includes dashboards, IoT Devices, data analytics, discovery, etc., as well as a corresponding Life Cycle. The proposed solution has been validated against a large number of IoT applications, as it can be verified by accessing the https://www.Snap4City.org portal; while only three of them have been described in the paper. In addition, the reported solution assessment has been carried out by a number of smart city experts. The work has been developed in the framework of the Select4Cities PCP (PreCommercial Procurement), funded by the European Commission as Snap4City platform.

## 1. Introduction

The concept of the Smart City is becoming increasingly important and pervasive, making it a major research area for businesses and policy makers [1]. Smart Cities are currently required to provide flexible architectures in order to satisfy new functional and non-functional requirements that arise and constantly evolve in many different contexts and environments. There is also the necessity of scaling up to handle the increasing number of users, devices, and services. However, the current common approach is yet to develop ad-hoc applications for specific domains, which leads to the production of non-interoperable services [2]. Smart Cities are becoming more focused on activities related to Internet of Things/Internet of Everything (IoT and IoE) every day. This has provoked a switch of technology and approach in field data collection. In the past, most city-generated data were collected from domain-specific vertical applications by sending data to dedicated concentrators (for example, the collection of data from traffic lights, traffic flow sensors, parking, etc.). Each and every vertical application used its own solution for data collection, either via radio or other means. On the other hand, the microservice architecture paradigm has arisen from the large-scale industry requirements of building and maintaining large-scale distributed systems [3], supporting an adequate availability, scalability, modularity, as well as flexibility [4]. The new push for IoT is stimulating cities to adopt the same gateways to collect data on multiple services. This means that the infrastructure of the sensor network has to be properly designed and planned by city operators. LoraWan may be an option [5], while in many cities, 5G solutions are quickly starting to replace former technologies, at least in large cities [6].

However, the arrival of complex IoT streams via 5G is not a problem for smart city data aggregators serving the Smart City Control Room [7]. Presently, they collect data from several heterogeneous sources, and to reduce the number of data concentrators to be aggregated would be a benefit, rather than a problem. On the other hand, the IoT/IoE paradigm is also strongly impacting infrastructure management [8,9], since the systematic adoption of IoT has also led to the adoption of the PUSH approach (event-driven protocols), which has brought forth the possibility of acting on actuators and not only sensing the city, but also creating Event Driven Applications. Event Driven Applications are also called IoT Applications, which have to be capable of processing messages and producing reactions in real time [10]. For many respects, Smart City applications constitute one of the most complex cyber physical systems, CPS, due to their complexity in terms of data, data analytics, and interfaces with the real world, both physical and digital user interfaces.

Concerning the state of the art, several different definitions and architectures have been proposed for IoT frameworks [11]. Most of the IoT platforms adopt a code-based development in the design of IoT Apps. Some of them provide development tools, such as Eclipse and JavaScript. Others provide simple tools, such as IFTTT [12], relying on rule-based scripts [13]. Other platforms adopt solutions similar to ETL (Extract Transform Load) processes [14], which are capable of event-driven processes, or just substitute it for high-rate polling. One of the most promising tools for creating IoT App is Node-RED [15], based on Node.JS, which has been proposed by the JS Foundation, as a fully open source application. The Node-RED approach is a mixed visual composition of nodes/blocks, which is used to compose the so-called flows that are concurrently executed by a Node.JS engine. It is quite diffused and also directly provided in official releases of IoT devices, such as those released by the Raspberry Pi family. It is provided with a minimum set of functionalities (the building blocks/nodes), while other blocks, loaded from a large library, which are made available by the JS Foundation, can be easily added. The classical nodes provided in the standard version can be classified as: input, output, function, social, storage, analysis, advanced, and dashboard. Such nodes are not sufficient for creating Smart City IoT Applications. In fact, in the context of a Smart City, they were not powerful enough to cope with the basic requirements of such a domain, even though their use in the field is quite widespread. In the next section, many other solutions have been considered and compared in order to select the starting point of our research and innovation action, i.e., the Snap4City platform, funded by the European Commission, as described below.

### 1.1. Related Work

Several Smart City use cases have been presented in the literature, transforming a collection of huge quantities of data related to urban environments into useful content for various stakeholders (citizens, tourists, local government, and companies) [16]. Among them, some of the most notable use cases concern the cities of Santander [17], Manchester, Barcelona [18], Singapore [19], Seoul, San Francisco [20], Rio de Janeiro [21] and Florence. However, most presented solutions still focus on specific domains, aiming at resolving particular problems, with little software reuse [22].

In order to manage the high variety of devices and applications, including IoT, mobile, web- and service-oriented frameworks, the Microservice architecture has been increasingly adopted in recent IoT-based solutions for Smart Cities. In fact, the Microservices architecture involves the development of simple and loosely coupled services [23], enhancing scalability and availability, facilitating maintenance and fast isolated testing. These aspects were incorporated with the aim of simplifying the complexity of the traditional service-oriented architecture (SOA) [24].

The programming of IoT Applications is performed in several different manners using different tools [25]. For example, in the AWS (Amazon Web Services) IoT ecosystem, a complex stack of Amazon’s services is proposed, which enables developers to define any business logic by programming the flow of IoT Devices’ data towards their visualization in a proprietary dashboard. IoT Devices can activate the AWS functions written in Java, Python, and C#. These functions can implement business logic or trigger actions in the IoT world or the Cloud. Data rendering is flexible and can be used for data inspection and reporting. Azure IoT provides a development environment for building business logics. Its integration with mobile applications is performed using the Xamarin platform in C# and NET. In addition, it is also possible to write applications in Java, Node.js and Python. For the creation of dashboards, developers can use the MS Business intelligence tool. In the Google IoT platform, several programming languages, like Java, Node.js, Python, Go, Ruby, PHP, and C#, can be exploited to program the data flows from devices to dashboards. In such cases, data flows are implemented using programming languages. None of them provide a visual editor for an IoT Application flow definition to be used, even without any software programming skills. In addition, these solutions often require the mandatory use of cloud services, and they are not suitable for any autonomous on-premise deployment scenario. As a limit case, they may allow for some modules to be configured on-premise via IoT Edge, with reduced capabilities, connected with the cloud solution.

In [26], a survey on Visual Programming Languages (VPL) for IoT was proposed. In the survey, 13 different VPLs were considered. The analysis mainly focused on comparing them on the basis of the programming environment, licensing, project repository, and supported platforms. Some of them are Open Source platforms, while others are proprietary. Among the Open Source platforms (Node-RED, NETLab, Ardublock, Scratch, Modkit, miniBloq, and NooDL), only some can be programmed using a Web interface and executed on some dockers or on-cloud virtual machines. In [27], the use of haptic interfaces in the context of IoT Applications for smart manufacturing was reviewed. The analysis produced a classification and the identification of the main elements needed to exploit haptic devices, without encountering the problems of microservices and VPL in programming IoT Applications.

In the assessment of Open Source VPL for IoT, several important factors should be considered, such as:**Performance** in terms of the number of messages processed per second and per minute;**Diffusion** of the tool, namely, the size of the communities supporting the tool, both as developers and users developing applications and additional features;**Openness in terms** of the flexibility to create new blocks in order to extend the basic functionalities, for example, exploiting external services, developing data analytics, etc. As another example, in the domain of a smart city or industry 4.0, you could add visual modules representing the IoT Device elements, as well as Data Analytics;**Cloud/docker support** for their on-cloud use and/or support for any execution on embedded IoT; thus, executability on several operating systems. Some of these VPLs produce only code for specific embedded devices. The automatic on-cloud resource provisioning and the elastic scaling of resources are needed to manage a large amount of IoT applications;**Level of expressivity**: an IoT VPL can be oriented toward defining the functional or data-driven/flow aspects. The first are typically less expressive, since such constructs are related to the programming language: section, assignment, condition, iteration, etc. Data Flow models are more expressive and may be easily extended with new blocks/functions, and this lack of expressivity may be solved if the right blocks are identified. Therefore, this may involve producing simple data flows, even for complex problems. In fact, the **complexity** of the graphs produced for solving problems in the applicative domain can be measured in terms of the number of blocks/modules and connections used to solve such problems;**Usability**, including the visual editing of data flows via a web browser or through specific client tools, etc. The fundamental features, such as inputs, outputs, function definitions, social gathering, storage load/save, data analytics and dashboards (user interfaces), may be not enough for creating complex Smart City IoT Applications with little effort. On the other hand, one may have a well-designed VPL, and yet the effort may be fruitless due to a lack of expressiveness and a domain suitable for MicroServices;**Managing resources**: the possibility of developing IoT applications (as well as portions of them) and data analytics to share them in a marketplace. In the context of VPL for IoT, the development of Data Analytics is quite complex, since processes may have a lot of dependencies, and the programmers’ skill may imply knowledge of the data storage and models, in addition to knowledge of the algorithms for data analysis. This approach may require the involvement of experts on sensors, algorithms, infrastructure, programming, knowledge modelling, domains, etc.

Some solutions that can be suitably used to address such factors and overcome the problems described above are, for example, Node-RE, which can be executed on multiple operating systems. It has a visual programming via a Web interface, and the source code can be loaded on several different embedded devices, etc. [15]). NETLab is an open source development environment for an embedded system [28]. Ardublock is an Open source block programming for Arduino, which is mainly a functional approach and is limited to specific devices [29]. Scratch is a visual MIT tool, adopted for code generation for IoT. It is a very low-level programming model and is not event-driven or functional. Modkit is strongly oriented toward Embedded IoT devices. miniBloq is a block functional programming at a very low level. NooDL allows for the fast prototyping of a user interface, and it is less suitable for data analytics. VISUINO is an open source tool for developing code for embedded IoT, but it provides limited capabilities for being used on-cloud. Kura is an Open Source Java visual tool for IoT. FLOGO is an Open Source tool for data flow in Go, which natively works on AWS Lambda [30]. Finally, Wyliodrin is a development environment for the deployment and updating of IoT edge embedded devices.

Most of the above solutions are strongly focused on producing code for embedded devices with a functional model and are not data driven/flows, for example, Scratch, Modkit, miniBloq, etc. Therefore, they are not suitable for creating IoT Applications that can be used with IoT Edge, as well as on-cloud docker containers, to address the complexity of Data-Driven applications and flows and exploit Data Analytics. Solutions, such as AWS, IoT Azure, and IoT Google, are more oriented toward traditional programming. The VPL tools that are data-driven, open to extensions, supported by large communities, and can be used on both edges may be few: Node-RED and maybe the emerging FLOGO. In [31], a Node-RED flow editor has been used to create smart city IoT applications for projects, such as VITAL http://vital-iot.eu/project. The suite of nodes implemented is not focused on the functional aspects of IoT Applications, so the resulting flows are still too much focused on the technical aspects and are not user-centered. In fact, as depicted in [31], even the creation of simple activities, such as the identification of traffic flow sensors, results in the development of complex workflows with a high number of nodes/blocks. Specialized workflow formalization models have also been proposed, for instance, in [32]. In [33], a Model-Driven Development (MDD) process has been proposed, which consists of (i) a tool for creating semantic algorithms, and (ii) a workflow generator that works on the basis of the first results. The user interface is a subset of the Node-RED environment.

There is a few IoT development environments, where algorithm repositories are provided for their exchange in the community, such as in [34] and [35]. This feature should be mandatory in cloud-based IoT development environments, such as Axeda [36], BlueMix [37], and ThingWorx [38]. On this last point, there are two levels in IoT valorization environments:IoT platform marketplaces, such as the BeeSmart City, Fi-WARE, EOSC, etc., which are typically constrained to a class of solutions or open to all of them. Their target is the promotion and marketing of IoT or Cloud solutions.IoT Applications and solution resource sharing and marketplaces, such as the Node-RED library, which are typically focused on specific IoT kinds of development processes and on the promotion of interchange.

In this paper, the issues related to the Snap4City IoT development environment and framework are addressed. The experience described herein refers to the design and implementation of the Snap4City platform, available at [39,40], which is based on Km4City [41]. Snap4City is the solution produced in response to a research challenge, launched by Select4Cities PCP (Pre Commercial Procurement), which was part of the H2020 research and development project of the European Commission [42]. Select4Cities has identified a large number of functional (mainly Smart City IoT) and non-functional requirements (open source, scalability, security, GDPR compliance (General Data protection Regulation), working on-cloud and on-premise, etc.), which are fully described on the web site and aim at creating the best solution for modern Smart Cities that supports IoT/IoE for public administrations and Living Labs. Most of the identified requirements have been taken from the large association of Living Lab ENOLL (European Network of Living Lab association, [43]), and from consultations with smart cities at the level of the European Commission, such as EIP-SCC (European Innovation Partnership on Smart Cities and Communities, [44]).

Snap4City was a response to the research challenge and could prove to satisfy all the Select4Cities requirements. Snap4City allows for the creation and management of user communities, which, collaboratively, (i) may create IoT Solutions, (ii) exploit open and private data with IoT/IoE, respecting GDPR, and (iii) create/use processes and IoT Applications that could run on Edge, Mobile and cloud, with the capability of interacting with one another and with users via messages, Dashboards and Applications. Special attention has been devoted to the creation of an integrated development environment for IoT Apps (cloud and edge), based on VPL, with dashboards, and supporting data analytics and resource sharing, thus minimizing any required technical expertise from programmers. To this end, a large number of visual elements have been identified and developed as MicroServices for the Smart City IoT. The starting point was Node-RED, of which we could identify many breaches to cover the Smart City domain. In addition, the developed platform has full support for users, providing assistance during the development life cycle, from data gathering to dashboard production, as well as in IoT App development, in the sharing of results with other developers and in the community, as self-assessment, together with security support, as described in [45].

It should be remarked that this paper presents information that has not been presented in our previous articles. In particular, the suite of MicroServices, their motivation and requirements in relation to smart city IoT solutions, a number of integrated examples, the Life Cycle of development, and the validation of the approach are presented.

### 1.2. Structure of the Paper

This paper is structured as follows. In Section 2, the requirements for a microservice-based programming solution for smart city applications are analyzed and reported. Section 3 presents the Snap4City Architecture, with a stress on the data flow, IoT and MicroService aspects. In Section 4, the Snap4City MicroServices Library is presented, and the motivations behind it are explained. Section 5 introduces some applications that exploit the MicroServices in the context of emergency management, personal mobility, and crowd monitoring with mobile PAX (people/person) Counters. In Section 6, the Snap4City development life cycle is formalized. Section 7 presents a validation of the solution, performed with city officials in developing IoT Applications. The conclusions are reported in Section 8.

## 2. Requirements for MicroServices-Based Smart City Applications

As described above, the aim of the research reported in this paper is to design and implement a visual programming environment, in which city operators may develop Smart City Applications with IoT by means of visual constructs. In this section, we identify the requirements of the challenge.

As a first step, in analyzing the state of the art, we realized that a VPL for developing IoT Applications has to provide **generic requirements,** so as to support:**Data communications:** Sending/receiving messages/ data, providing both push and pull modalities. This means that the language has to be capable of performing requests using many different available protocols [46], such as pull protocols to obtain data (e.g., Rest Call, Web services, FTP, HTTP/HTTPS, etc.) and push protocols to receive data via data-driven subscriptions (WS, MQTT, NGSI, COAP, AMQP, etc.). In the context of IoT solutions, this feature is quite common. For example, in the Node-RED library, you can find a large collection of nodes covering tens of protocols, while large platforms, such as AWS and IoT Azure, are typically limited and address just a few protocols. In some cases, such limitations concern the supported security and authentication schemas, since in most cases, a username and password (or authentication keys) have to be included directly into the flow in clear text. Thus, some approaches to IoT security are unsatisfactory [41]. For example, in Node-RED, each single IoT Device can be connected to a flow using specific credentials, which are directly included in the code. On the other hand, a generic service for IoT Discovery and Registration is missing.**Data save and retrieval** to/from some bigdata storage. For example, the capability of storing data and exploiting them to implement algorithms for predictions, decision-making processes, value trends overviews, etc.**Data transformation and processing** via the algorithm formalization and the possibility to start, pause, and recall the data execution periodically or sporadically, according to the arrival of some event or other firing conditions.**Calls of external services** in order to enable the instrument of delegation to external solutions, when it comes to computing and processing, for instance, the computation of some complex Data Analytic algorithms or the exploitation of external services, such as the Social Media Analysis via REST Call. For this purpose, the classic Node-RED provides access to Watson IBM for machine learning and artificial intelligence reasoning. Desirable applications could be, for example, computing sentiment analysis, performing a clustering classification and recognition, computing some predictions on the basis of the historical data of some City IoT values.The **presentation of data** via some Dashboards that would constitute the user interface of the application. Data visualization should be expressive enough, including values in real time, time trends, histograms, pie charts, etc.The uses of the **Dashboard** as **a user actuator interface, enable the user to act**: (i) on some internal variables and (ii) to send messages/commands to IoT Devices and flows.

The VPL IoT platforms should present a number of **non-functional requirements,** such as demonstrating the capabilities of robustness (in terms of availability and fault tolerance), scalability (to be capable of serving from small to very large businesses with corresponding volumes of processes per second), security (authentication, authorization, secure connection, etc.), full privacy respect (compliance with data privacy, according to the GDPR, General Data Protection Regulation European guidelines), interoperability (e.g., communicating with any kind of protocols, devices, and services), and openness in terms of being an open source and having the possibility of adding new modules and functionalities, etc. On the other hand, most IoT development environments are not web-based development platforms and are focused only on generating code for specific embedded platforms, such as Arduino (see above), VISUINO, Ardublock, ModKIT, miniBloq, etc.

In addition, by analyzing the requirements identified by Select4Cities, ENOLL, EIP, etc., we realized that a large number of **specific**
**smart city IoT requirements** have to be satisfied by a VPL to enable the easy development of IoT Applications in the Smart City context at the service of City Operators and advanced City Users (citizens, stakeholders, third-party developers of SME, researchers on data science, etc.). For example, the VPL should provide specific features to address the functionalities of developing applications in domains, such as **mobility, environment, parking, culture, health, tourism, energy, and general city services domains**, with no need for knowledge of technical details about the provided services or service supplier identities. In fact, in almost all cities, different mobility, environment, tourism, energy, education services, etc., are provided by different operators. Moreover, some of these services are also related to one another, for instance, the number of free parking lots and the events in the area, the number of weather forecasts and the number of people in city gardens, the traffic flow and the environmental data regarding NOX, etc. That is, when it is supposed to rain, the number of cars in traffic increases, and thus the number of free parking slots at certain hours decreases.

On this basis, a number of **specific functionalities** have been identified to develop smart city applications via VPLs, which should be available in terms of MicroServices and building blocks, in order to:**Access Smart City data entities** on the basis of their functional semantic descriptors, for example, the status or prediction of “*parking square Carlo IV*”, rather than the variable “Status” of the device, “sensor45”.**Discover and exploit city entities’ data**, regardless of their technical details, such as their push/pull protocols and gathering model, message format, provider, etc., while focusing the search on the functional aspects and ignoring all the details about the query syntax used in the specific storage (e.g., SPARQL, SQL, and ElasticSearch). For example, the applications should provide answers to the following questions: “*let me access all temperature values in this area, path or close to a given GPS point”*; or “*give me the sensors measuring PM10* (or any other environmental variables) *close to my position or along a path”*, regardless of the actual connection of the sensor devices, which would be connected to the platform using different IoT brokers, different protocols, etc.**Create a Graphic User Interface for web and mobiles**, with the visual representation of complex data in dashboards, also allowing for the input of data into IoT Applications and the whole system. Node-RED provides a number of basic nodes to create GUI for IoT Apps (Dashboard nodes), whereas the offered solution presents strong limitations regarding its usage in the context of a smart city. They do not present maps (with pins, paths, areas, etc.), heatmaps, trajectories, origin destination matrices, trend comparisons, Kiviats, multi-pie charts, etc. Furthermore, they are not protected by an authentication and secure connection. In smart city domains, advanced data types need to be managed and shown, for instance, maps, paths on maps, heatmaps, collective trajectories, area shapes on maps, areas, ortho-maps, origin destination matrices, etc. Most smart city data types are related to sequences of GPS points.**Access to Data Analytics and their production.** The real need, in the context of a Smart City, is to grant data analysts with the possibility of creating some data analytic processes (which may be in R, Tensor Flow, Python, Java, etc.) and using them in data flows as MicroServices, with no need for a programmer to step in, nor for an administrator to take action every time. This approach may create a strong flexibility in IoT Applications. For example, classical Data Analysis algorithms for smart cities could include solutions for:
○**Computing the routing from point A to point B according to certain criteria**, for example, using a car, public transportation, a bike or travelling by foot; the shortest or fastest; and avoiding specific zones due to road works or other kinds of emergencies. This implies the production of a sequence of points connected by street segments.○**Collecting and computing metrics on social media data from Twitter** (such as counting Tweets, retweets, as well as extracting Natural Language Processing features [47], i.e., verbs, adjectives, citations, hashtags, and performing sentiment analysis on them), for example, computing metrics related to Tweets collected on the basis of a set of keys, Twitter usernames, hashtags, etc.**Register new IoT Devices,** regardless of their protocol, which are provided via some external or internal IoT Broker. This feature is strongly relevant to the registration of personal devices. For example, a glucose meter, security devices for home monitoring, etc. The data collected from those devices have to be kept strictly private and in accordance with the GDPR guidelines [41,48].**Save and retrieve personal data**, for example, data coming from private devices, such as those mentioned in the previous points, and thus according to GDPR.**Load and publish Data Sets**, aiming at ingesting open data and/or creating new open data sets and sharing them with other cities and networks. A data set is typically a file in some open format that contains a data description, for example, the position of benches in the city, the position and a description of all the restaurants in the city, etc.**Provide different kinds of events within the city**. They can be (i) notifications of consolidated events (such as road accidents and/or fire brigade and police calls), (ii) entertainment and sport events, or (iii) potential emergency calls, i.e., events that are not yet consolidated, which may be produced by police or citizens in the street to communicate facts (for example, potholes in the street, a balcony with some problems), etc. They need to be profiled with a number of metadata, and in the case of emergency events, they should be compliant with the CAP standard (Common Alerting Protocol) [49], in order to provide tools and instruments to enhance a city’s resilience capabilities, and be interoperable among several operators, such as the fire brigade, local and national police, civil protection, etc.

## 3. Snap4City Architecture

In this section, the general architecture of Snap4City is presented to contextualize the successive sections, and the VPL aspects are discussed. The Snap4City platform has a Lambda architecture, as shown in Figure 1. Data are received in push and/or pull, and almost every datum can be considered an IoT data source. They may come from IoT Brokers, social media, web server, web sockets, streams, etc.

In Snap4City, most data provided from External Services or operators are collected in pull via REST Calls using scheduled processes, written in ETL, through the Penthao Kettle tool. The ETL processes are executed (periodically or sporadically/on demand) on a cluster of Virtual Machines by a scalable and distributed scheduler, called DISCES [40]. The collected data are regularized/reconciled, according to the Km4City Knowledge Base [40], and then pushed into the Big Data Cluster storage. The reconciliation process allows for the reconnection of data to city entities already in place, for example, by connecting POI (Points of Interests) to civic numbers, traffic flow to street segments, etc. The aim is to create a fully connected knowledge base for the city, where semantic queries can be performed by exploiting physical and conceptual relationships.

On the other hand, data arriving in push (data-driven) are typically real-time data streams, managed by a number of IoT Brokers, which the Apache NIFI distributed cluster is subscribed to. The NIFI process is in charge of performing a regularization/reconciliation task on the data, according to the Km4City Knowledge Base, and it then pushes the resulting data into the Big Data Cluster storage, while indexing and thus creating the Data Shadow. The above-described parallel solution tends to be normalized in a single approach, based on NIFI [50], when the platform is small. Otherwise, when the platform has large data volumes, a distributed data warehouse for data ingestion, based on ETL, is more effective. The Big Data cluster storage includes an HDFS Hbase, a Phoenix storage, an RDF store for semantic reasoning, based on Km4City (which is operatively implemented on the Virtuoso store), and an ElasticSearch Index. The whole solution presents textual indexes on all fields, semantic indexes, and elastic aggregations to perform advanced “faceted” filtering queries. The indexing and query supports are exploited by Snap4City Smart City APIs [40]. When small installations are performed, the HDFS cluster can be avoided.

The operative IoT processes (IoT Applications) can be executed on-cloud or on-premise. In Figure 1, the cloud case (private or public) is depicted. When IoT Applications are executed on IoT Edge, which is located on the field, it may directly communicate with the IoT Applications or Dashboards on-cloud or by means of IoT Brokers (to which all of the others can be subscribed). On such grounds, without the risk of losing generality, in Snap4City, the IoT Applications for Smart Cities can be obtained as follows:IoT App = Node-RED + Snap4City MicroServices.(1)

The IoT App exploits the basic nodes of Node-RED Node.JS plus Snap4City MicroServices, which are suitable for smart city data transformation and processing.

The Node-RED platform is based on two components: (1) a web-based visual editor to design flows and (2) a runtime environment that may execute flows. In Snap4City, the Node-RED editor has been improved to:Communicate with the so-called Resource Manager,Save and load IoT Applications and manage SSO (Single Sign On) with Snap4City,Manage a large set of MicroServices, namely, the Snap4City Libraries of Nodes, which are accessible from the Node-RED official library.

Moreover, the runtime engine of Node-RED has also been improved to (i) manage the security, according to SSO and the Snap4City model, and (ii) execute it on Docker, according to the elastic management solution of Snap4City.

On the other hand, the changes performed on Node-RED have been released as open source and are functional only for large-scale on-cloud use, while the Snap4City library can be used in the installation of IoT Edge with the standard Node-RED tools, without any restrictions.

The Snap4City solution has formalized and implemented a large set of MicroServices, satisfying the above discussed requirements. The MicroServices provide an easy and formalized access to all the Smart City services that are available on-cloud from the platform (including the ones to control a part of the platform itself). They are made available in the Node-RED Node.JS environment to create IoT Applications as VPL. Among the MicroServices, the IoT applications also need to access such services to allow for the exploitation of Data Analytics, Visual Analytics and Dashboards. The latter two aspects can be employed to create the Graphic User Interface (GUI) of the IoT Applications. These tools, orchestrated by the IoT Application flows, may automatically inform, announce, act and produce alerts and warnings on IoT Devices, networks, the user interface, external services, etc., and provide support to close the loop towards the user acting/reacting on the GUI and/or Devices, including notifications.

In the deployment of the Smart City IoT as a Service SCIaaS, such as Snap4City, and in large smart city applications, a relevant number of IoT Applications may need to be deployed on the basis of the on-line requests, made by users/organizations. Snap4City has recently been accepted by the EOSC (European Open Science Cloud) marketplace of the European Commission, and the IoT Applications therefore need to be managed and allocated on-demand on the basis of the users requesting them, as well as be executed on-cloud. To this end, an elastic (vertical and horizontal) infrastructure has been created to manage—in a scalable manner—the IoT Applications in containers [51] and mechanisms to guarantee end-to-end security [41]. As shown in Figure 2, the user interface allows the user to manage its own IoT Applications, irrespective of whether they are on-cloud IoT Apps or IoT Edge/field Apps, whether they are child processes/containers for Data Analytics or WebScraping, etc. Please note that different kinds of IoT Applications are represented by different icons.

The Snap4City MicroServices abstract low-level details for the programmer using a visual environment, hiding the complexity of the sophisticated algorithms and tools. This is useful and suitable, for example, for providing routing, a spatio-temporal search and discovery, for data analytics, dashboarding, networking among IoT devices, IoT data abstraction, etc. The Snap4City MicroServices are distributed into two official libraries of Node-RED nodes by the JS Foundation portal). The two libraries are dedicated to final users (basic) and to developers (advanced). The version dedicated to Users provides outputs of Node-RED nodes that can easily be exploited by non-skilled users on JSON. In fact, most of the output produces single variables and not complex JSON-structured messages. On the other hand, the version for Developers (to be installed on top of the basic version for final users) presents a number of nodes/blocks that can accept complex JSON message inputs to create strongly dynamic IoT Applications.

Both Libraries of Snap4City Nodes can be directly installed in any Node-RED tool on any operating system: Linux, Raspberry pi, Windows, etc. In addition, we have also developed an Android App that executes Node.JS/Node-RED and our libraries to allow for the use of them on IoT Edge and also exploit the mobile device sensors on the above-mentioned operating systems. In [51], we demonstrated how the Snap4City approach may work on mobility and transport applications, where critical safety communications and solutions have to be set up, involving IoT networks with IoT Devices and IoT Edge, IoT Apps and Dashboards.

In this paper, a deep view of the analysis and design of Snap4City MicroServices for smart city applications is presented. The analysis has also been supported by the evidence on how the MicroServices can be successfully used for IoT Application implementation, so that they can be used by City Operators and Final Users.

## 4. Snap4City Library of the MicroServices of Smart Cities

In order to satisfy the smart city requirements, reported and discussed in Section 2, in Snap4City, a collection of more than 150 MicroServices, as Nodes for the Node-RED programming environment, was developed. The Node-RED philosophy of visual programming allows for the creation of event-driven data flow applications, where the exchanged messages are in JSON format. On the other hand, periodic processes can also be developed by scheduling one or more internal timers. This means that users can develop IoT Applications as Node-RED flows, exploiting both Push and Pull data protocols, in the same visual programming environment. In the context of smart cities, both protocols are needed, while IoT Applications have to be capable of creating flows and exploiting a large number of features that are typically not available in the Node-RED open library, nor in a number of libraries from different providers. Moreover, the Snap4City MicroServices are at a level that can allow even non-expert users to easily develop IoT Applications for smart cities, which is assessed in Section 6.

The most relevant families of nodes/MicroServices for smart cities are listed below, and they perform different kinds of activities, which are useful in the IoT App construction.

**Access to Smart City Entities**, which have different data models and, thus, different MicroServices may be required. Some Entities may have simple sets of Metadata, for example, the ones describing the POI, e.g., the title, description, web, email, GPS location, images, opening time, etc.; others may have complex information and/or specific data types and structures, for example:Status of the first aid: number of people under observation for each color in the triage, waiting time, etc.;Bus stops: all the bus lines, including the geometry, their planned time schedule, real-time delays, etc.;Sensors, along with their values, measurement units, types, healthiness criteria, etc.;Weather forecast associated with an area/region, which consists of a large set of values: temperature, humidity, pressure, wind, etc., for many different time slots in advance;Shape of cycling paths, gardens, parks, difficulties, restrictions, etc.;Parking areas, with the number of free spaces, predictions, typical daily trends in free spaces, costs, etc.;Events: (i) entertainment, with their description, photo, start date, end date, etc.; (ii) police officers on the street; (iii) emergency events, such as a civil protection early warning, according to the CAP standard, etc.

In order to simplify this complexity, MicroServices, like “*Service Info*” and/or “*Service Info Dev*”, are provided for Final Users and Developers, respectively. In the IoT Application, a search/Discovery has to be performed, as described in the next paragraph; otherwise, the developer needs to know the so-called ServiceURI, which is the unique identifier of all of the city entities in the Km4City Smart City Knowledge Base. In the Snap4City development environment, the ServiceURIs can be recovered directly from the graphic configuration of the MicroService or searched using the ServiceMap visual tool, which provides a GUI for queries. This means that the IoT App programming is 100% visual, even if a single service is used to access a single element.

**Search/Discovery of City Entities** and their relationships. The search of city data has to allow users to discover data/device values by a semantic search using a composition of the available query types, which are as follows:**Semantic classification**. In Snap4City, all the POI and Services are classified into more than 20 classes (mobility, energy, banking, environment, cultural activities, etc.), including a total of more than 520 subclasses (see, for example, the menu of the ServiceMap reported in Figure 3);**Geo spatial references**: close to a point, max. distance from a given point, along a path, into an area/polyline;**Textual keyword substrings**: for example, on the basis of the title and descriptions of city entities;**Value Types**: for example, all sensors measuring the temperature, all bus paths, etc.;**Historical time slot**: for example, data values for the last 7 days;**Prediction time slot**: for example, data regarding the predicted parking slots for the next 15 min, 30 min, 1 h, etc.;etc.

The results of this kind of search can be a single element, with its description, as well as a JSON containing a list of entities. In this latter case, the list can be split into single messages using the Node-RED Split node. Once the list of ServiceURIs is accessible, their detailed description can be obtained using the above-mentioned “*Service Info*” node. The search facility for Final Users is provided via the Node interface, simply performed by employing a user interface for setting parameters. On the contrary, as for the nodes for Developers, search parameters can be also prepared and sent to the search Node in JSON. In both cases, the user does not need to know any query language (e.g., SQL or MySQL), and he/she does not need to know if the data are coming from a complex set of queries on SPARQL, for the RDF store, Elasticsearch and Phoenix/HBASE. Therefore, the developer of the IoT App does not need to know all the tiny details about the large variety of adopted storages. Snap4City provides more than 70 different nodes for searching different Smart City entities, providing results in different data types: POIs, time trends, values, events, schedules of buses, bus lines, recommendations, addresses, routes, etc. This approach significantly simplifies the creation of Smart City Applications.
**Discovering and Exploiting IoT Entities (sensors and actuators)** should not be different from discovering any smart city entity. In Snap4City IoT Devices, data values can be accessed, searched and discovered by the above-presented MicroServices/nodes. On the other hand, when the user/developer would like to create a data-driven IoT Application exploiting specific IoT Devices, some specific MicroServices can be used so as to discover the desired devices, regardless of the IoT Broker, IoT protocol or IoT Device. To this end, the IoT Directory MicroService exploits the services of the Km4City Knowledge Base for searching, managing and discovering all of the available IoT sensors/actuators [40]. In Snap4City, developers can register on the IoT Directory, IoT Devices and IoT Brokers, supporting a large number of different protocols and authentication models [41,48]. The IoT Directory automatically registers new Devices in the Knowledge Base, and each new IoT element receives a ServiceURI, thus becoming a City Entity and POI [48]. A number of IoT Brokers are provided by Snap4City, while others are managed by third parties, for example, to provide an answer to queries, such as “give me all temperature sensors close to my house” (regardless of the data providers, protocol, source, etc.).**Creation of an advanced Graphic User Interface**, including graphics widgets, such as Dashboards, Virtual Devices that act on the IoT Applications and IoT Devices, advanced tools, etc. This means giving developers the possibility to design the IoT App user interface of the IoT App. The user interface has to show and also allow for interactions between users through messages in the IoT Network, including the IoT App, which may implement the logic of the user interface. The user interface is built by composing a number of graphics widgets in a connected Dashboard to present and collect data. In Snap4City, the IoT Apps may be connected with Dashboards by means of widgets for:
**Rendering data**: single content, time trends, bars, histograms, maps, pies, semaphores, dynamic signals, buttons, knobs, clocks, etc.**Collecting data** as a switch, a knob or a keypad, which are interactive elements on Dashboards and are represented in IoT Applications as Input Nodes for the flow and are able to provoke events in the IoT App, according to the data-driven approach.**Showing MicroApplications and External Services** in a generic iframe widget.

In addition, other Dashboard Widgets may not have a counterpart in the IoT App and may be directly added to the Dashboard using the Dashboard Builder editor [52], which can:include **Visual Analytics** tools, such as External Services, MicroApplications, object Tracking, Origin Destination tools, heatmaps, traffic flows, maps of any kind, etc.**Visualize data** from other sources: IoT Brokers/ Devices, data stores, API, etc.include **Virtual IoT Devices** that can send data directly to IoT Brokers, as an IoT Device but from the user interface.

This also means that an IoT Application may be connected to multiple Dashboards, and a Dashboard may be connected with multiple IoT Devices and IoT Apps [7].

**Data Analytics,** which would mean providing MicroServices/nodes that allow a number of data analytic services to be exploited in the IoT Applications (for example developed in R, See Figure 4). In Snap4City, it is possible to: (a) exploit the computing of data analytic processes that are already in place, (b) develop new data analytic processes, (c) call External Services, as a rest call. In fact, the Snap4City suite provides MicroServices that implement:**Data Analytics** for computing: routing, auto-ARIMA predictions, anomaly detection, descriptive statistics, machine learning predictions, heatmaps, providing social media data from Twitter Vigilance [53], etc.**A generic Data Analytic MicroService,** where Data Analytic algorithms, developed in R-Studio, can be executed, according to specific guidelines.**A REST call invoking External Services**, for example, gaining access to Twitter Vigilance API, The Weather Channel API, The Things Network API, etc.
**Save and Retrieve Personal Data,** for instance, time series for motion tracking, values of personal devices, clicks on mobile Applications, POI, shapes, KPI (key performance index), Keys to access IoT Device services, etc. The possibility of saving and retrieving data from a safe storage (with the possibility of assigning delegations according to GDPR) enables a large variety of smart scenarios for the final users and operators, for example, saving personal data from personal health devices (e.g., monitoring glucose), from home statuses, from the location of mobile phones that belong to a specific person. In addition, the Snap4City platform automatically collects all of the personal data gathered from the mobile Apps accessed on the Snap4City login (provided that the user authorizes their collection with signed consent, according to GDPR).**Save, retrieve, and publish Data Sets**, as those managed in the open data portals. Most public administrations publish their data sets, in the form of open data, and also share their data via federated networks (see, for example, the harvesting mechanisms of CKAN [54]). Having the possibility of creating data sets from the flow of IoT Applications means that city operators can automatize their ingestion/update processes and their production/publication.

In the Snap4City suite of MicroServices, there is also a number of tools for managing the DISCES backoffice scheduler of processes, as well as for saving LOGS of data accesses and flows. The former can only be accessed by administrators, while the latter can be useful for both developers and administrators. Moreover, a number of so-called GEO-utilities have been identified and implemented in Snap4City, such as MicroServices/nodes to:Calculate the distance from two GPS locations.Verify, once a GPS point and a closed shape/area are given, if the point is inside the area or not in order to, for instance, verify the match with administrative areas and thus determine whether your dog is in the garden or the monitored bear is in the forest, etc.Obtain the most probable value of a variable represented in a Heatmap from any GPS point, irrespective of the presence or absence of a sensor at that point. This feature is very useful in choosing among different routes: the quietest one, the less polluted one, the one that has the least traffic, the busiest road for meeting people, etc.Obtain the closest civic number and street from a GPS point or vice-versa. This is very useful for geo reversing.Obtain the closest road map segment (typically called a node in Open Street Map language) from a GPS point or vice-versa. This can be useful for routing and computing precise distances, and it is very useful for geo reversing.

This means that IoT Application developers do not need to solve the so-called direct or inverse georeferencing problems, since they are provided by default.

A number of Node-RED nodes that are useful in smart city contexts are also available: they have not been implemented by us, as they are globally provided by third parties, for instance, interaction via Facebook social media, SMS, email, Telegram, etc.

## 5. Example of Application

In this section, three examples are presented, with the aim of depicting the capabilities and the expressiveness of both the IoT Apps and the Snap4City development environment. They are IoT Applications for: (i) emergency management, (ii) routing on the basis of environmental data, and (iii) the dynamic management of PAXCounters IoT devices for monitoring people flows at museums and events.

### 5.1. Alerts about Critical Events Involving the People in a Specific Area

In this section, we present a scenario in which a public operator (Road Operator) on the field, like a policeman or a public transport driver, notifies a control room operator (City Operator) of a critical event in the city. The notification includes a report in real time of the event position, the number of involved people and the seriousness of the event.

The City Operator would like to: monitor the events and services in the city, receive critical event notifications from Road Operators and assess the contextual conditions and services status. The control room receives the notifications and can explore the status of the services in the city, evaluate the gravity of the situation and, therefore, take the correct decisions to cope with the event. Especially if many events occur in the same area, a critical condition may be detected early, and an Early Warning may be propagated by other police officers. The Road Operator would like to monitor the status of traffic, parking, environmental variables, speed limits, services and send critical event notifications via coded descriptions. Figure 5 better explains this scenario.

Figure 6 provides an overview of the City Operator’s Dashboard, named “The City Operator”. The City Operator can see the City Map (1) and select one or more services on the selector on the left (2) to see them on the map. By clicking on the elements forming part of the list on the left (2), the services belonging to the selected category are displayed (or hidden) on the map. Multiple categories can be selected at the same time to gain a broader overview of what the situation is like in the area where the Road Operator is. By clicking a POI on the map (1), a pop up is shown that represents the real-time data of the selected POI (if available). The data shown in the pop up can also be represented as real-time values in a dedicated widget (3), as well as time trend values in another specific widget (4). When the Road Operator sends the message related to a critical event, the City Operator receives the notification in the widget (5), which lists all of the notifications received. The minimap (6) is automatically centered on the coordinates sent by the Road Operator.

The creation of the City Operator dashboard is performed using a wizard that guides the user in the choice of variables that he/she wants to monitor and insert into the dashboard, as well as the widgets that will display the values produced by such variables [7]. Please note that Widget (5) is a data-driven visualization from data derived from the IoT Orion Broker in NGSI.

Please note that the two Dashboards shown in Figure 5 have been created in two different manners, according to the:**Dashboard Wizard,** which permits, through simple guided steps, the selection of data sources, then the decision as to which widgets will be used to visualize them, and finally, the dashboard creation and the creation of the selected widgets in a few clicks.**IoT Application editor** (Node-RED), selecting Dashboard MicroServices and connecting real-time data to allow them to automatically appear in the dashboard as Widgets.
Please note that both approaches are valid and can be mixed.

When the **Dashboard Wizard** is used, as in Figure 6, the user is requested to choose one of the available templates and the structure that the new dashboard is to be based on. If the user chooses a particular template, it is also possible to subsequently add more widgets manually or by calling the wizard again within the newly created dashboard. In the Wizard, the user can choose the data sources to be used inside the dashboard. The types of data sources are various: A Complex Event, IoTApp, External Service, Heatmap, MicroApplication, My Personal Data, MyData, KPI, MyKPI, POI, MyPOI, Sensor, Actuator, Special Widget, or WFS. Multiple data sources can be selected and displayed inside the same dashboard. The data generated by the selected sources can be very different from one another, and the user must choose how to represent them within the dashboard. In the Wizard, for each selected datum, the user may choose among the possible widgets that are to represent and display the specific data type. Therefore, the possible graphic widgets are dynamically shown. To make the choice easier for the user, when he/she has selected a certain data source, widgets are automatically filtered, and vice versa. The user could also select the widget model for the dashboard first, and then the wizard will filter the data sources, showing only those that may be visualized on the selected widget type.

Once the new dashboard is created, if the user wants to add more widgets, the dashboard can be edited through the wizard again and again. The adopted approach allows the user to easily create interactive dashboards, without using any particular programming skills, relying only on the help of the wizard. The Road Operator’s dashboard has been created using the **IoT Application editor**, which is the second approach, listed above. The Dashboard Wizard is not used to create this dashboard. In fact, it has been created in the automatic mode from a Node-Red stream. To allow the IoT App flows to interact with the Dashboards, we have to create Nodes/Blocks, representing Dashboard Widgets, in the flow. Widgets in the flows can be INPUT widgets (providing data from the Dashboard to the IoT App) as well as OUTPUT widgets (graphically representing some data on the Dashboard). These elements can also be regarded as Virtual IoT devices, i.e., Virtual Sensors and Virtual Actuators, respectively.

The Snap4City Dashboard nodes allow for the creation of 5 types of widgets on the dashboards among a range of available ones. These widgets will represent the different metrics chosen by the user/developer/city operator and are as follows: A Gauge Chart, Speedometer, Time Trend, Single Content and Web Content. The first 3 types accept numerical values as inputs. In the first two types, the last sent value of the represented metric is displayed, and in the third type, the metric history (as a time graph of all of the metric values sent over a period of time) is also shown. The Single Content can receive data in any text format and can also display HTML. The WebContent allows the user to send a url and display a resource that is located at that url within the widget.

The nodes creating Actuators on the Dashboards are of 5 different types: Impulse Button, Switch, Numeric Keyboard, Knob and Geolocation. The Impulse Button sends an “On” signal, when pressed, and an “Off” signal, when released. The Switch allows the user to set the button to “On”, until its status is changed. The Numeric Keyboard allows the user to send any numeric data from the dashboard to the stream, showing the last value sent in the widget. The Knob allows the user to dynamically change the sent value, depending on how the knob is turned. The Geolocation sends the position retrieved from the browser, where the dashboard is shown.

In Figure 7, the relationships between the User Interface of the Road Operator and the corresponding IoT Application are shown. The Node-RED flow is divided into three parts, which are described below. Figure 7b shows the GPS Position: the buttons receive the GPS Position, which has been set to “Helsinki”, “Florence”, “Antwerp“, and “My position” in this example. By clicking the GPS position button, the map on the top sets the position accordingly. The flow makes the request to the Smart City API, and the map is shown in the right place with the list of services and PINs on the map. Figure 7c shows the People Number: The numeric keypad Widget allows the Road Operator to insert the number of people involved in a critical event. The user enters the number and presses the “Confirm” button. The confirmed number is shown in the “Last confirmed” box on the top-right of the numpad. In the IoT App, the number is stored in a temporary memory.

Figure 8 shows the Alert Color: the colored buttons allow a color code of the emergency to be set. If the user has correctly set the GPS position and inserted the number of people involved, by pressing the color code, he/she can also send the message to the City Operator by posting it on an IoT Virtual Device on the Dashboard.

When the Road Operator presses the color button, indicating the priority of the emergency, a JSON is created, containing information about all of the previously described aspects, such as the number of people, the coordinates and the color of the emergency. This JSON is sent as a value of an entity created on a Fi-Ware Orion Broker through a node, which is enhanced with the ability to add authentication keys, if compared to the base node, provided in Node-Red. This ensures that writing such an entity can occur only if the IoT App, which is sending the data, belongs to the entity owner.

The entity registered on the IoT Orion Broker can also be chosen in the wizard, which allows the user to create the widget that can be seen in the City Operator’s dashboard at the top right (5). This widget adapts the displayed value dynamically when a new JSON arrives, thus changing the state of the entity without the City Operator needing to update anything on the dashboard in order to comply with the Event-Driven messages. The flexibility of the dashboards allows for the creation of mixed dashboards containing widgets created with the wizard and widgets automatically created by the Node-Red streams. If the dashboard is created automatically by the creation of widgets from a NodeRed stream, the user can open the wizard and add additional widgets through the different modalities, explained above. The messages exchanged between the Node-Red streams and the IoT Orion Broker use the NGSI protocol, which is the same protocol used when values are received from the widgets on the dashboards linked to the real IoT Devices of the IoT Orion Broker. The exchange of messages among the nodes dedicated to the dashboards (whether they are actuators or viewers) is conducted via Web Socket Secure, which is protected by an Access Token, depending on the owner of the same IoT Application.

The additional nodes, created to enhance the functionality of Node-Red in the Smart City, can also be installed on local users’ devices (such as Raspberry). In some cases, it is necessary to request and obtain an account on the Snap4city platform, before exploiting such enhanced functionality. Without this account, the user cannot obtain the Access Token that ensures the security of the communication among nodes and other components of the Snap4city platform [41].

### 5.2. Check Which Route is Less Polluted

The above examples have shown how it is possible to use Snap4City MicroServices for the development of systems useful for the security and control of emergencies in the public sector. The suite of microservices used in the above exercises have been mainly composed of dashboard elements. The IoT App with a dashboard can also be used by citizens in private applications. In the example developed in this section, MicroServices retrieve information from the Smart City storage and info to create a dashboard that provides the user with pieces of information on which of several jogging paths is the least polluted at a precise moment. If predictive data are available, it can make predictions.

In Figure 9, the dashboard presents two possible jogging paths, and the values of O3, PM_2.5_ and PM_10_ of the two paths are displayed in the center.

The flow realizing the logic behind the dashboard is shown in Figure 10. This dashboard was created automatically, when the Node-Red flow was deployed for the first time. In the flow, the outermost blue nodes on the right correspond to the two maps with the routes, the six central blue nodes on the right are related to the Gauge Charts, reporting the values of the pollutants, and the only blue node on the left corresponds to the “Update” button, which allows the data to be recalculated, when pressed. These nodes automatically create the widgets on the dashboard, if the dashboard exists; otherwise, they also create the dashboard, without any required intervention by the user, who has to edit only the name of the dashboard and of the Widgets, when configuring the nodes.

The orange nodes on the left, called “PARCO DELLE CASCINE” and “PARCO VIA FONTALLERTA”, allow the user to calculate the shortest path between two geo-points. With a simple visual configuration, as shown in Figure 11, you can choose a starting point, an arrival point, the time you want to begin the trip and the transport means. The node will produce a JSON, containing all of the information about the calculated route, in the output. As can be seen, the complexity of the route calculation is transparent to the user, who can view the result of his request in a simple and intuitive way.

In this Node-Red flow, unlike in the previous example, a minimum programming ability is required. The JSON that is returned from the previous nodes also contains the complete path, as calculated by the system, and this must be divided into its main points, exploiting the function nodes, made available by Node-Red (where the user can interact with the variables by writing code in JavaScript).

The other six orange nodes on the left, which are internal to those just described and have the names of the pollutants, allow the user to obtain the value of that substance at a given geographical point in a timely manner.

In this case too, the complexity lying behind this value is hidden to the user, who manages to obtain that value in just few steps, thus being able to use it as he/she pleases. The complexity behind the whole described scenario, which is transparent to the user, implies the retrieval of all of the sensor data available in the area under consideration, the calculation of a heatmap (furthermore, interpolating values in the points where there are no sensors), its storage on a geo-server, and finally, the timely recovery of the data available from the same geo-server, along with the creation of APIs dedicated to this particular service. The calculation of the heatmap must be conducted periodically, and in turn, this periodicity has been conducted using a special Node-Red flow that executes (on the pre-determined frequencies or scheduled time intervals) the R script in charge of calculating the heatmap. Please note that the sensors are typically located at some specific points, which may be very distant from the computed routing. A specific service in the smart city back office is dedicated to computing the heatmap and to providing a MicroService, which allows the value of a pollutant, as well as of the temperature, humidity, etc. at any point of the area to be obtained.

In the flow depicted in Figure 10, the values of the polluting elements are calculated for the main points of the routes returned by the first two nodes, and the average is calculated, so as to obtain an indicator of the pollution for that route. The user who has created a dashboard, made in this way, can press the update button, before going jogging, and he/she will know where to go running in order to breathe cleaner air. The automated update can be programmed as well.

### 5.3. Controlling the Personal Mobile PAX Counter

In the first example, we have seen how to create two interoperable dashboards through an IoT App in order to ensure support for emergencies within a city. We have also demonstrated how this allows for smart interaction with Road Operators, who are located directly at the emergency scene, as well as with City Operators, who are located inside the City Control Room. In the second example, we have shown how to create a dashboard that retrieves data from city sensors to obtain real-time information on the state of the air quality. We have also stressed how this can be useful for decision support, for instance, when making choices based on pieces of information offered by the system, and how it is able to improve the user’s health by suggesting the most virtuous behavior. In both cases, everything was conducted through MicroServices, and the user did not need any additional device to obtain the required information. All of the temporary data for the processes have not been permanently stored.

This example depicts the interaction with IoT Devices in counting people using Wi-Fi and Bluetooth sniffing in the vicinity of the device (according to GDPR [55]). In Figure 12, a mobile PAX Counter is shown, based on ESP32, sending messages to Snap4City via LoraWAN. Other simpler versions can be located at fixed places and may also exploit WiFi to send the measures obtained also from different, additional sensors. In the current example, they have been used to monitor a Museum in Antwerp and many other locations. A total of 22 devices have been installed. The measured values are sent to the LoraWan operator, The Things Network, which does not provide a system for data analysis, except for a way to receive and manage devices.

## 6. Museum Case

In the literature, there are several technologies for tracking the movements in a museum and understanding the inside flows [56]. On the other hand, their costs are very high, and for most applications, obtaining the internal traces is unnecessary. In the presented IoT Application and Museum case, with free entrance, the monitoring of the flows of visitors entering and exiting the museum, collecting also the history of such data and not only the real-time values using low-cost solutions, has been of great interest. In fact, monitoring via the PAX Counter of 10 doors would cost less than 3000 euro. To this end, data have to be saved permanently into a time series data store, for example, in MyKPI and MyPOI storage via MicroServices. The Snap4City approach allows IoT Developers to create and retrieve data entity trends directly in a Dashboard and for Data Analytics, as data Shadows. According to this solution, an IoT App has been created for the PaxCounter to receive the new data coming from The Things Network in Event-Driven mode (as IoT Gateway), according to the Publish Subscribe model of MQTT; then, such data are saved as MyKPI via MicroServices. Consequently, the Dashboard Wizard automatically sees the data entity on the MyKPI and allows us to automatically create Widgets, showing their trends, real-time values, etc. In fact, the Snap4City back end provides a suite of scheduled scripts, specifically designed to automatically and programmatically populate the Dashboard Wizard with data from all the different data sources (Sensors, IoT App, External Services, Heatmaps, Personal Data, KPI, MyKPI, POI, MyPOI, etc.). Figure 13 shows the Dashboard, presenting the time trends of the entrances and exits of the museum and all of the derived data. Such derived values of the initial real-time data can be computed in real time, saved on a personal storage and shown on a dashboard via the IoT Application, exploiting Snap4City MicroServices. At the same time, while sums are being calculated, the difference between people entering the museum and those leaving is also performed in order to obtain the number of people inside the museum in another entity.

Through the Dashboard widget and IoT App, it is possible to send commands to the Pax Counter via the IoT Gateway in order to obtain the status of a device, as well as to change settings. A command that can be useful when using a Pax Counter installed in the museum can be changing the mode of counting people from cyclic to cumulative. In the cyclic mode, the count is reset every 8 min, and the anonymous data of the devices, counted by the Pax Counter, are deleted. Therefore, in the following 8 min, if a person stands near the Pax Counter, that person is counted again. In the cumulative mode, the count is not reset, but is rather increased with new people arriving near the Pax Counter. Therefore, duplicate counts of the same person are avoided, until the device is in the cumulative mode.

## 7. Mobile PaxCounter for Monitoring Events

The mobile PAXCounter, presented in Figure 12, has been used to monitor sporting and entertainment events. Each new measured count is associated with GPS coordinates and sent to the platform, thus resulting in trajectories with associated values, as depicted in Figure 14. This is conducted using the Snap4City Tracker Widget.

In this case, as already shown above, it could be very important to change the modality, allowing for switching from Cyclic to Cumulative counting. To this end, in order to program the change of modality, an IoT Application with a dedicated Dashboard user interface has been realized, as reported in Figure 15.

In this case, the IoT Application is more complex, since it requires (i) collecting data; (ii) programming the IoT devices; (iii) interacting with the user to collect the data for programming purposes; (iv) monitoring the status of the IoT device to verify if the setting has been successfully applied; and (v) reporting on the status of the IoT device to the user. All these functionalities have been implemented in a single IoT App, as reported in Figure 16.

Thus, a custom widget has also been developed to allow for the definition of both the beginning and ending instances of the programmed period in the cumulative mode. The activation of the programmed interval is performed by pressing the “Activate” button, which saves the times within the IoT App. The IoT App performs the change of modality at the correct time, since the IoT Devices are not programmable. In addition, 3 Single Content widgets are instantiated to display the activated period in the cumulative mode, the status of IoT App for sending data to the device, and the state of the IoT Device. In Snap4City, Custom widgets can be developed by creating them in SVG and implementing a specific process to connect them to Snap4City MyKPI.

Once the custom widget has been created, the IoT App is created, realizing the control logic covering all the functionalities, listed above, by exploiting the MicroServices for: A dashboard widget, personal storage exploitation, IoT Gateway connection, etc.

From the point of view of the life cycle, the above presented IoT Applications can be saved into the Resources Manager, thus allowing other users to download and install them in their own resource area and use them with their own devices.

## 8. Development Life Cycle of Snap4City Applications

The design and development of IoT Cyber Physical System/Solutions, CPS, is much more complex than regular software development approaches in terms of the SW life cycle. An aspect that they share is the adoption of strongly dynamic microcycles. In [57], a review of the life cycles for IoT has been presented. The life cycle proposed is a simplification with respect to actual needs, since it does not take into account data analytics, personal/private data, and complex dashboards. In most cases, when IoT development life cycles are discussed, the focus is on device development, finalization, testing, deployment, connection, etc., while in [58], the complexity of the life cycle is much greater.

The Life Cycle for the IoT Applications/Solutions in complex domains, such as a Smart City, involves a number of specialized tools, such as those for Maps, Dashboards, etc., which need to be well placed in the life cycle. Therefore, in this section, the development life cycle for Snap4City applications is presented, as shown in Figure 17. In the represented life cycle, only the software aspects are addressed.

In the context of Snap4City IoT Smart City development, the life cycle should start with the Analysis and Design to produce the requirements and specification of the application. This information includes the following:
**Data** which has to be identified on the basis of the goal of the scenarios, particularly the real-time inputs and outputs that could be used; Historical data that could be exploited to show them or for Data Analytics; data to be scraped from Web pages, according to the terms of use; and private information of the users or entities involved, according to the GDPR guidelines. This process of data analysis may also imply the Discovery of Data on the available knowledge base via a Service Map or Data Inspector. If the required data are not available, a specific ingestion process will be developed.**Functional transformations** of data and commands, which may be required on the basis of the accessible data format. This activity could also create a business logic of the IoT Application, taking into account historical data, inputs, MyKPI, scraped data and Data Analytics to produce outputs on other devices and/or dashboards.**Data Analytics** to be developed for the application, if needed. For example, for Smart Parking, a free parking slot prediction will be needed, and this is evident from the analysis. The **Data Analytics** will have to be developed from scratch or customized, exploiting some ready-to-use MicroServices.**A visual Interface**. In most cases, the graphic interface is part of the problem. A good visual representation may make the difference between a tedious view and a nice user-centric view. This part includes the design of the Visual Interface in terms of the Widget and Dashboard. The single elements may present data from the data store/shadow, may be directly produced in/for the IoT Application data flow in real time, or may need to be developed as Custom Widgets.

Once the Analysis and Design has been completed, a global view of the Smart Application is sketched, but it still needs to be implemented. Thus, the different aspects and phases may be addressed by different tools and for large factories also by different teams, addressing:**Data Ingestion and Modeling**. In this case, different tools can be used.
○**An IoT Directory** for registering new IoT Devices, IoT Brokers, etc. In some cases, the definition of the IoT Device Model is needed to pass through the IoT Device Registration in Bulk.○**A DataGate**/**CKAN** for the ingestion of Open Data with geo information, such as POI.○**ETL** development, a process of the ingestion of some files or data that have to be taken in PULL, for example, from WS, FTP, HTTP, etc. They could even be developed in the IoT Application tool, exploiting MicroServices.○**External Services** for the direct identification of external API that could be exploited in the IoT Application.○**MyKPI** for the modeling and registration of the KPI and data needed to obtain permanent and personal data in the IoT Application.**Development of Data Analytics**. This process is typically addressed by Data Analysts, skilled in data science, accessing historical and real time data and creating specific algorithms, such as prediction, anomaly detection, interpolation, clustering, etc., exploiting libraries and tools of operative research, statistics, machine learning, and using languages and the development environment provided by Snap4City for online development, such as RStudio, Python, Java, etc. The Data Analytics may produce:
○**MicroServices** that can be reused in a range of IoT Applications. They produce data on Storage to be visualized on Dashboards by regular widgets, such as Heatmaps.○**Special Tools** that may produce views, such as External Services on Widgets, for example, Origin Destination Matrices. Additionally, these Special Tools may produce data in the Big Data Storage,**Data Flow Development**, that is, the development of the **IoT Applications**. Data flow development is typically focused on producing outputs in storage, MyKPI and/or presenting resulting data on some Dashboard Widgets. This IoT Application development exploits the Snap4City library of MicroServices/Nodes, plus those that may be newly created for the specific solution. The development of some IoT Applications has been presented in the previous section. The IoT Application implements the business logic of the solution and can be deployed (on-cloud/container or on IoT Edge). The Deployment can be performed in a sandbox for testing. When the IoT Applications includes the usage of Dashboard Widgets, they are created on the Dashboard at the moment of the Deployment of the IoT App.**User Interface Development**, which consists in the development of the **Dashboard**(s). The Snap4City Dashboards can be composed by several kinds of Widgets, mainly classified as:
○**IoT App Widgets**, which are directly produced or have a counterpart representation in the IoT Application (also called virtual sensors and virtual actuators). These widgets are used to manage event-driven graphic elements, as inputs and outputs of the IoT Application, such as buttons, messages, switches, gauges, data views, trends, keypads, semaphore statuses, etc.○**Custom Widgets**, which can be produced using SVG tools and may implement synoptics, as specific interaction tools, with animations, etc.○**Regular Widgets**, taken from the large collection of data viewers of the Snap4City Dashboard Builder.

The final result consists of a number of data ingestion processes, data analytic processes and MicroServices, IoT Applications, and Dashboards connected to each other. The Life Cycle, presented above in Figure 17, distinguishes between phases, formal documents, tools and processes, MicroServices/Nodes and the Widgets of the Dashboards.

The process of the Development, deployment, testing, validation, and acceptance testing is typically iterative, arriving at the final deployment in production with acceptance testing. Then, after the solution passes to maintenance, it has to be continuously monitored to detect eventual problems that may spring from missing data, a change of data format, exceptions in the data analytics or in the data flow, etc. All these sources of fault should be managed using a correct design and implementation, with error management and fault detection. The best solutions try to produce a resulting service to address, for example, the lack of data for making predictions via machine learning, providing the average values, etc.

## 9. Assessment and Experimental Data

In the literature, several Smart City experimentation testbeds have been presented, expressing the common requirement of involving citizens, ICT operators and local administrations. This implies the organization and deployment of experimentation facilities, such as crowdsensing campaigns, living labs and tools to monitor and enhance the engagement of citizens [59]. In order to assess the effectiveness of the Snap4City IoT Application Development Environment, a training course has been organized with Smart City operators, ICT city officials, and City Council Operators (who have never used Node-RED before). During the course, we assigned a number of exercises [60] and recorded all of the activities. Before executing the exercises, a specific but short training was performed on Node-RED programming and Snap4City MicroServices. Then, the exercises were presented, and a fixed time slot was assigned to learners to produce the IoT App. In both cases, the IoT also included the development of a small Dashboard. At the end of the exercise, we closed the sections, asked the users to provide some answers through a written questionnaire, while the IoT App and dashboards were saved on the platform and closed for assessment. Thus, the assessment considered both results, as described below:(1)The answer in the paper about the exercise, where users in the audience should mark the needed Node-RED smart city blocks, required to solve the problem. The form included a list of the NODES/MicroServices that can be used to compose the IoT App.(2)The developed of an IoT Application in the platform, accessible for our review, with the possibility of testing to verify whether it was 100% complete or not.

Please note that the assessment considered the fact that the results would not be unique, in the sense that each exercise could be solved in different manners.

The aim was to answer to the questions: “As for the training received from us, was it enough to allow users to develop an IoT App to solve the problem?”, “how much faster is the solution with respect to other solutions?”, “how has it been perceived?”, “has the user already learnt the programming model?”, etc. The audience included 24 skilled users, less than 20% of whom were ICT developers. The following exercises were proposed:Create an IoT App that reads a value from a Snap4City service (for example, the parking lot seen in the previous demo, identified by the serviceUri: http://www.disit.org/km4city/resource/CarParkPieracciniMeyer), and realize a flow that sends different messages (e.g., “Almost Full Parking slots” or “Free Parking”) on the dashboard, on the basis of a certain threshold. Apart from these functionalities, the creation of a service, which sends to you an email, containing ☺!, would be considered as a plus. (Time required: 15 min).Create an IoT Application/flow that: (i) reads a value from a list of Snap4city services (for example, the car parks in the Florence City Area, as seen in the previous demo), (ii) calculates the average of Free Parking Lots, and (iii) sends the value on a dashboard, with the four possible nodes seen in the demo. (Time required: 20 min).

The assessment was carefully performed, since each exercise could be solved with multiple solutions. To assess the exercise on the paper questionnaire, we only verified if the test-takers selected the right microservices. A score of 100% was given if they selected all of the right answers, according to the possible solutions, while a fraction of 100% was assigned proportionally, if some answers were missed. In the assessment of the final IoT App, we verified if the App satisfies the requirements, and a vote was conducted, according to the percentage of such satisfactory outcomes. In some cases, for example, when the developer added more features (for example, email sending), we positively considered this a plus (see Table 1).

As can be seen, from the results summarized in Table 1, 21 users performed Exercise 1 on paper. The average score on the paper assessment was 71.42%, with a variance of 302.8. This means that the mean value of the scores, assigned on the basis of the paper assessment, was 71.42%, which corresponds to the percentage of the identified Node-RED blocks needed to solve the problem. In addition, 20 of those 21 actually implemented an IoT App. Only one test-taker presented the result on paper, and he was not interested in performing the exercise on the real tool. The test-takers obtained an average score, in the real results, in terms of the coverage of requirements that was equal to 85.75%, with a Variance of 692.8. On average, the developed applications covered 85.75% of the requirements. In addition, 80.00% of users realized the IoT App, satisfying at least 75% of requirements. That is, in synthesis, a good result, confirming that the training was enough to let them create the IoT App autonomously. The test-takers who solved the exercise in the IoT App editor actually performed better with respect to the VPL tool, as compared with their results on paper. This also means that they could identify the blocks (the MicroServices), but it also stresses that the development environment provided a concrete support in correctly identifying the nodes/MicroServices by means of the over functionality, which provides a help description, when the mouse passes over the icons.

The 24 participants were also provided with additional questionnaires to assess their appreciation and effectiveness in a more general sense. In this case, we used a Likert scale of 5 values:There were 91% shown to be **satisfied with the training day**, among which 53.40% were very satisfiedExactly, 85% ranked the **IoT solution and Data Analysis as very good,** among which 42.17% ranked them as excellentIn total, 100% stated that the solution would be **useful for their work**, and 78.13% declared that it would be very useful in their daily workThere were 95.28% who found the creation of **IoT App somewhat easy**, among which more than 56% found it easy and very easyIn total, 93% described the functions for the **IoT App** (MicroService) creation as **complete** and satisfactory, among which 31.25% evaluated it as very completeExactly, 50% of users stated that the IoT App development was 5 times faster, compared to what they know is possible using other state-of-the-art tools.

Therefore, the answers to our initial questions have been obtained. A short training was sufficient to learn how to use the Snap4City IoT App development environment, and it is typically considered to be faster than any other tools (typically, they used ETL, programming languages, and rule-based solutions). The implementation of the applications that we presented, using other solutions or even the Node-RED standard version, is somehow unfeasible, because the complexity of some of our MicroServices hide, for example, the routing, the picker for the heatmap, the discovery, the dashboard, the IoT registration and discovery, etc.

## 10. Conclusions

Smart Cities are approaching the IoT world. Most of the early first-generation smart city solutions were based on ETL processes, which mainly support pull protocols and data silos. The last generation of smart cities often use IoT solutions and are thus moving towards the use of push protocols/event-driven applications. Therefore, the concept of Event-Driven IoT Applications is becoming more widespread and viable with respect to ETL or to rule-based solutions. In this paper, these aspects have been reviewed, highlighting the requirements for smart city IoT Applications, as well as those specific to the implementation of MicroServices for IoT Applications in Smart City frameworks. In this paper, we have also presented the Snap4City library of MicroServices for Node-RED, which has enabled the creation of a wide range of innovative IoT Applications for Smart Cities, involving Dashboards, IoT Devices, GDPR-compliant personal storage, geographical routing, geo-utilities, data analytics, etc. The proposed solution also addressed the formalization of the development life cycle, which we have adopted and proposed to develop a Cyber Physical System in Snap4City. The proposed solution has been validated against a large number of IoT Applications for the cities of Firenze, Antwerp and Helsinki. In addition, three applications have been reported and described in the paper, providing evidence of the relationships among MicroServices, Dashboards and data. Besides, the assessment of the solution has been performed and reported. The assessment has shown that the suite of MicroServices could make the development time significantly shorter, and it has been strongly appreciated by expert users. The research has been developed in the context of the Select4Cities PCP of the European Commission, as the Snap4City platform.

## Figures and Tables

**Figure 1 sensors-19-04798-f001:**
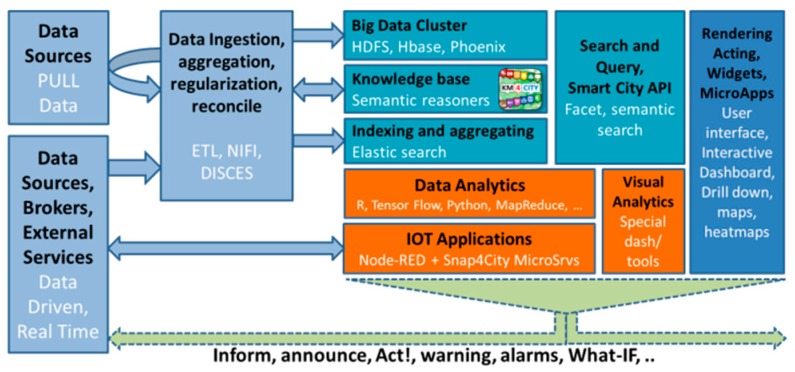
Cloud view of the Snap4City Functional Architecture, where the Internet of Things (IoT) Applications are only on-cloud.

**Figure 2 sensors-19-04798-f002:**
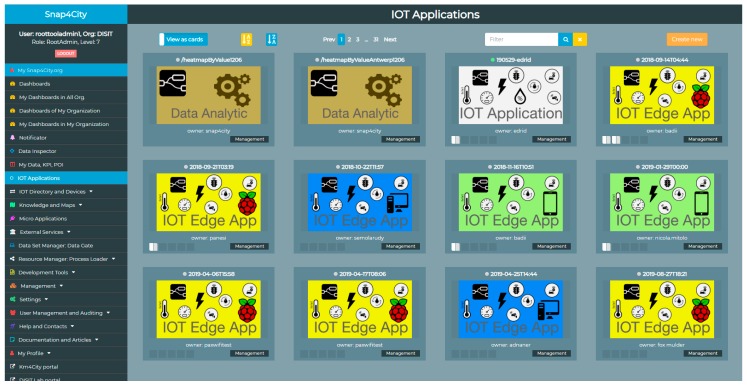
Snap4City: The IoT Applications manager, as seen by the user.

**Figure 3 sensors-19-04798-f003:**
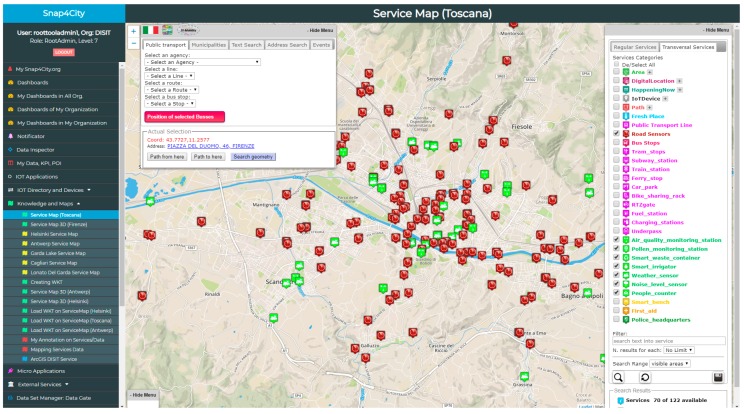
ServiceMap for ServiceURI identification.

**Figure 4 sensors-19-04798-f004:**
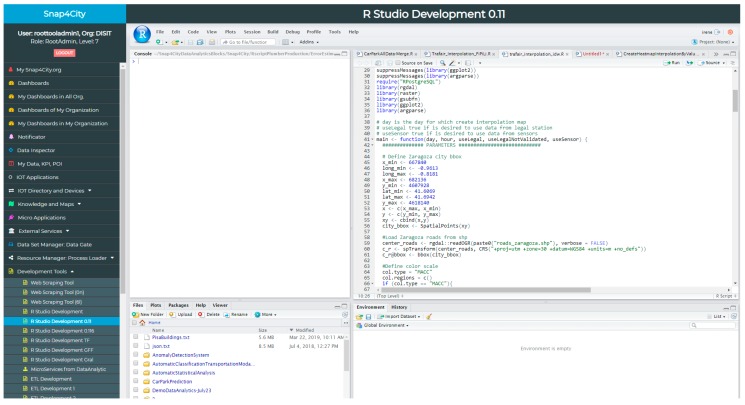
Developing Data Analytic MicroServices via R Studio to exploit them from IoT Apps in Node-RED.

**Figure 5 sensors-19-04798-f005:**
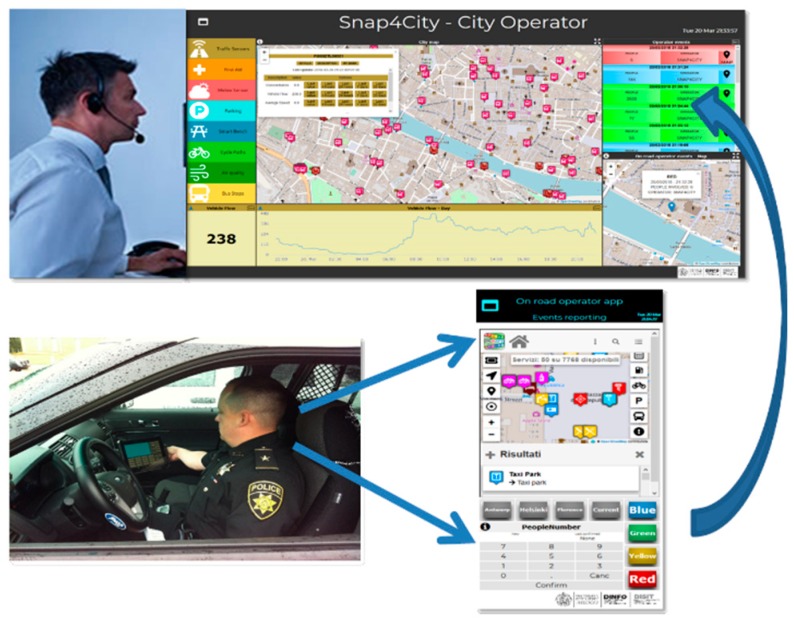
Connection between the City Operator dashboard in the control room and the one used by Road Operators.

**Figure 6 sensors-19-04798-f006:**
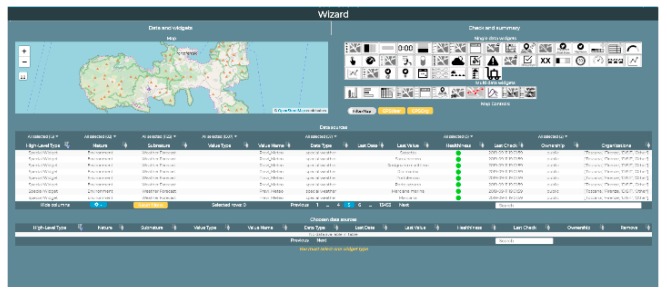
Dashboard Wizard.

**Figure 7 sensors-19-04798-f007:**
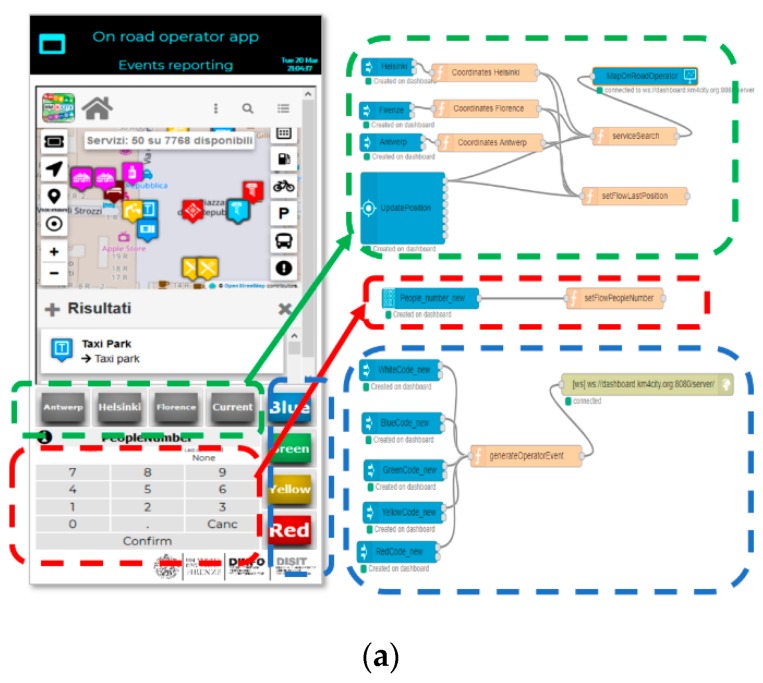
The Road Operator’s user interface and the IoT Application managing the logic. Overview and first details. Read the description on the text. Figure section (**a**) describes the relationships between the IOT App segments and the graphic interaction elements in the dashboard; (**b**) reports the logic for changing the area of work including the GPS capture; (**c**) includes the capture logic for the key pad.

**Figure 8 sensors-19-04798-f008:**
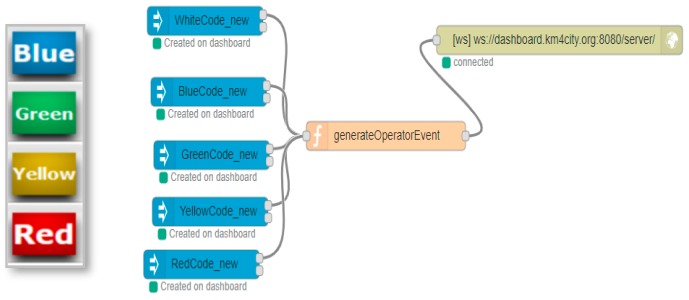
The Road Operator’s user interface, last part.

**Figure 9 sensors-19-04798-f009:**
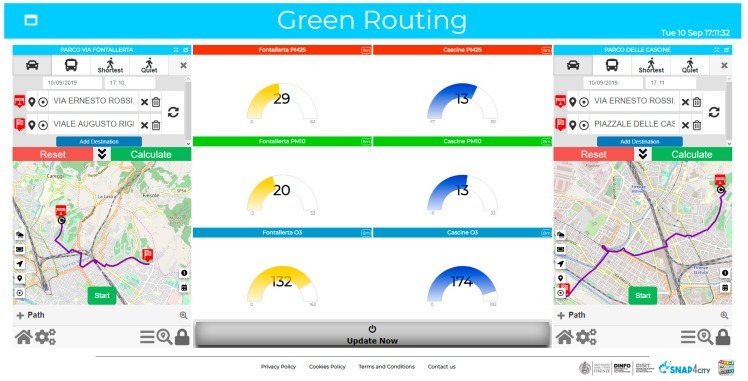
Personal Dashboard to decide the least polluted path for jogging.

**Figure 10 sensors-19-04798-f010:**
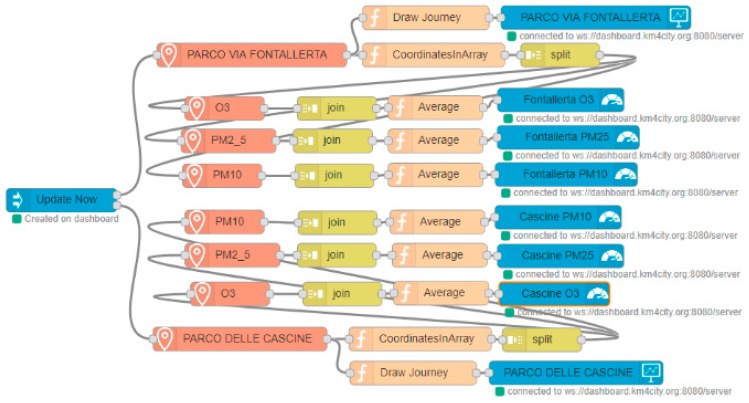
IoT Application, according to the logic of the Dashboard, as shown in Figure 9.

**Figure 11 sensors-19-04798-f011:**
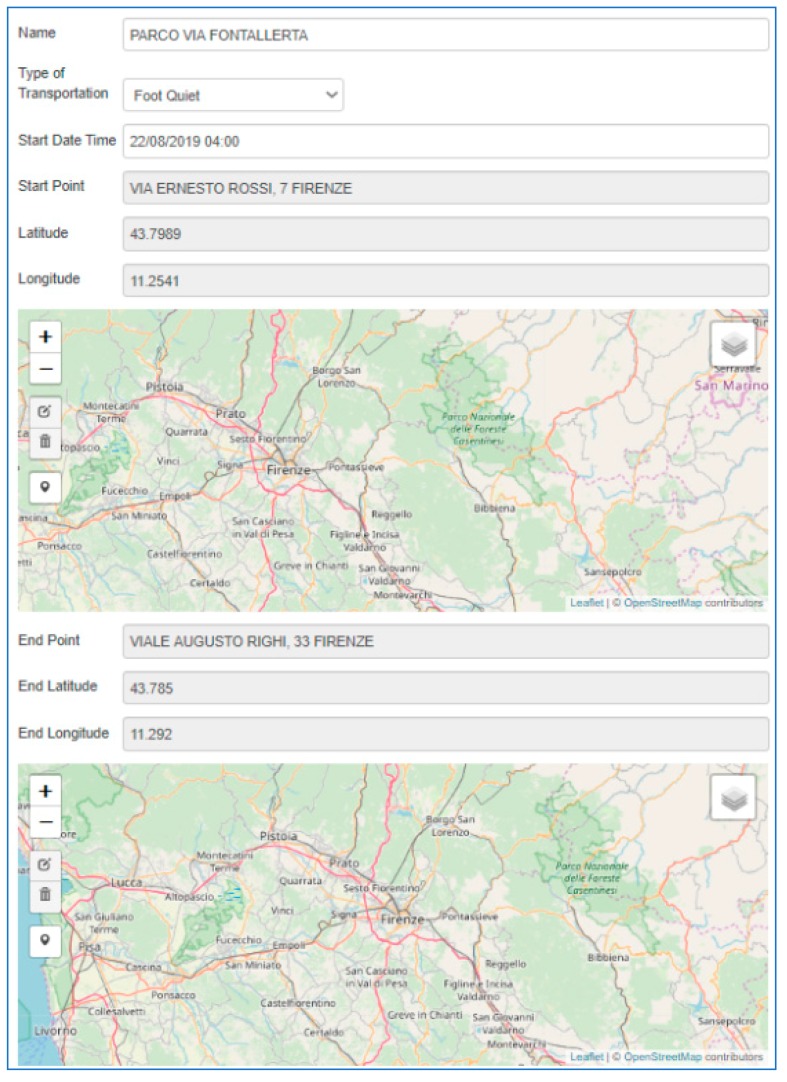
Setting for routing of the IoT App, as shown in Figure 10, and of the Dashboard, as shown in Figure 9.

**Figure 12 sensors-19-04798-f012:**
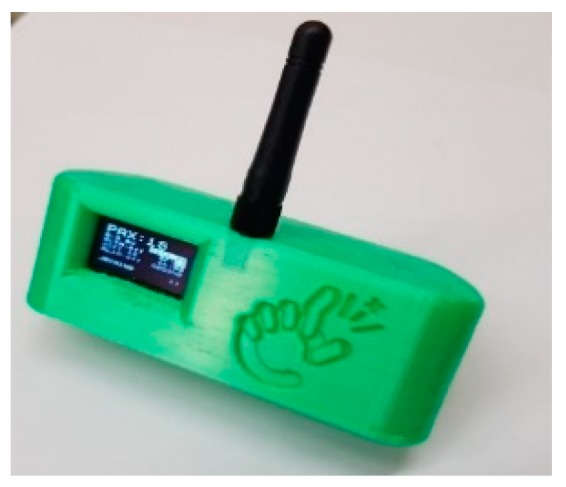
A mobile PAX counter based on ESP32.

**Figure 13 sensors-19-04798-f013:**
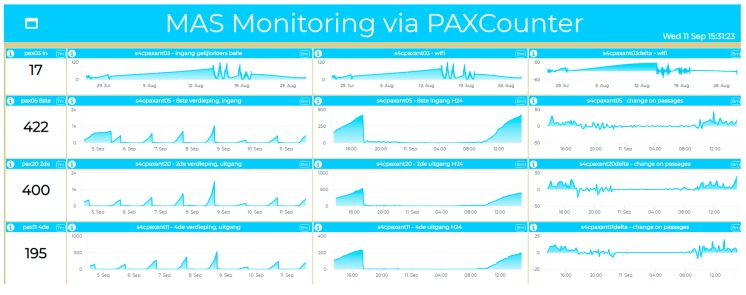
Monitoring the MAS Museum people flows.

**Figure 14 sensors-19-04798-f014:**
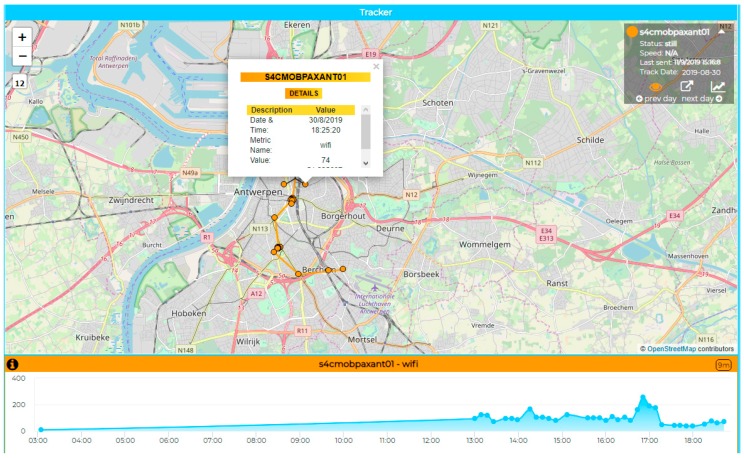
Tracking PAX Counters, measures over time and space.

**Figure 15 sensors-19-04798-f015:**
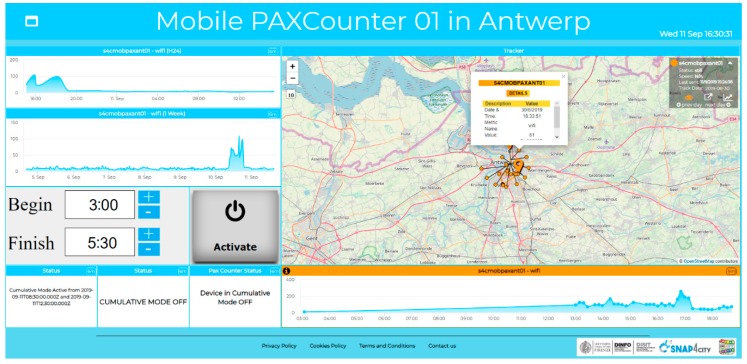
Management of the Mobile PAXCounter, with the possibility of programming a change of modality.

**Figure 16 sensors-19-04798-f016:**
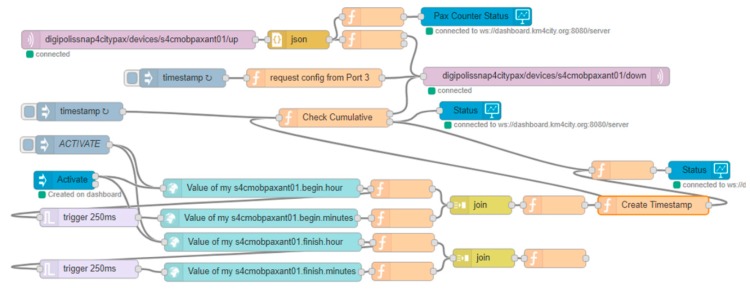
IoT App controlling the mobile PAXCounter.

**Figure 17 sensors-19-04798-f017:**
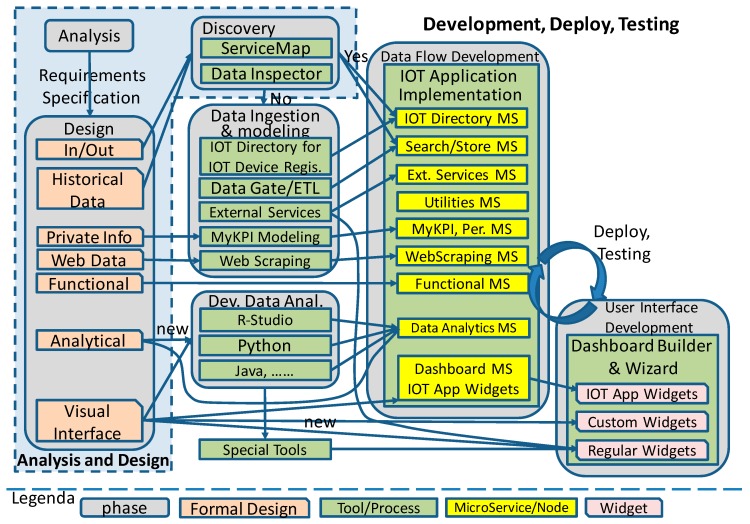
Snap4City Development Life Cycle for Cyber Physical Solutions, distinguishing phases, formal documents, tools and processes, MicroServices/Nodes and the Widgets of the Dashboards.

**Table 1 sensors-19-04798-t001:** Results of the IoT App Development Assessment.

**-**	**Exercise 1**	**Exercise 2**
Number Of Users	24	24
Exercise Duration	15 min	20 min
**Results On Paper**	**21**	**21**
Average Score	71.42%	83.92%
Variance	302.8	470.9
**Resulting IoT Apps**	**20**	**16**
Average Score	85.75%	83.42%
Variance	692.8	449.0
75% Of Requirements	80.00%	81.25%

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
