# Peer review of "MicroServices Suite for Smart City Applications"

_sensors, 2019, doi:10.3390/s19214798_

Round 1
Reviewer 1 Report
The paper describes a very interesting research project carried out by the authors, including a proposed tool and its assessment via an experiment with potential expert users of the tool. The assessment provided promising results.
The topic is very relevant and timely. The goal of providing an easy-to-use visual programming environment for city operators to develop smart city applications deserves attention.
However, the paper is not well written. The text is confusing in many parts and there are hundreds of grammar mistakes throughout the paper. Thus, the paper would need to pass through a comprehensive revision before publication. Both grammar and style would need to be revised.
Coverage of related work is weak. Most of the bibliography is composed of self-citations and references to underlying technologies, with almost no papers about Smart Cities, which was supposed to be a key concept of the paper. In particular, literature research on Smart City Platforms and Microservices-based solutions for smart cities should be included as a way to better compare the work carried out by the authors with the state-of-the-art in the field.
It could be useful to better explain the steps that a user should follow to create applications on top of the provided platform.
I like the assessment that has been done with potential users of the technology.
Finally, it would be important to make clear what is the original contribution presented in this paper and what has been presented in previous papers by the same authors.
Suggestion related to the project web site and not to the paper: overall the design of the site and of the tools is very poor. I'd suggest adding a professional designer to the team.
Author Response
The paper describes a very interesting research project carried out by the authors, including a proposed tool and its assessment via an experiment with potential expert users of the tool. The assessment provided promising results.
The topic is very relevant and timely. The goal of providing an easy-to-use visual programming environment for city operators to develop smart city applications deserves attention.
Answer: Thanks!
However, the paper is not well written. The text is confusing in many parts and there are hundreds of grammar mistakes throughout the paper. Thus, the paper would need to pass through a comprehensive revision before publication. Both grammar and style would need to be revised.
Answer: The paper has been extensively revised removing the grammar mistakes and improving the language style.
Coverage of related work is weak. Most of the bibliography is composed of self-citations and references to underlying technologies, with almost no papers about Smart Cities, which was supposed to be a key concept of the paper. In particular, literature research on Smart City Platforms and Microservices-based solutions for smart cities should be included as a way to better compare the work carried out by the authors with the state-of-the-art in the field.
Answer: We have strongly improved the reported state of the art in the introduction and in the related work, regarding both Smart City platforms and Microservices-based solutions, although the usage of microservices for Smart City is not so diffuse in literature as one may suppose.
It could be useful to better explain the steps that a user should follow to create applications on top of the provided platform.
Answer: In the new version of the article a new section 6 has been included presenting the life cycle for Smart City IOT Applications on the basis of our experience and tools.
I like the assessment that has been done with potential users of the technology.
Answer: Thanks!
Finally, it would be important to make clear what is the original contribution presented in this paper and what has been presented in previous papers by the same authors.
Answer: This has been clarified into the paper. It should be remarked that this paper is presenting aspects that have not been presented in our past articles in the past, and in particular: the suite of MicroServices, their motivation and requirements for smart city IOT solutions, a number of integrated examples, the Life Cycle of development, and the validation of the approach.
Suggestion related to the project web site and not to the paper: overall the design of the site and of the tools is very poor. I'd suggest adding a professional designer to the team.
Answer: Thanks a lot for the suggestion on the web site.
Reviewer 2 Report
This research is very interesting and adequate to this Journal, but will be improve using a correct design of experiments as in:
Application of IoT with haptics interface in the smart manufacturing industry
An considering your case in the museum, is very important describe a danger situation as in:
Innovative Data Visualization of Collisions in a Human Stampede Occurred in a Religious Event using Multiagent Systems.
A multivariable analysis is important in this research.
Stefano Valtolina, Fatmeh Hachem, Barbara Rita Barricelli, Elefelious Getachew Belay, Sara Bonfitto, Marco Mesiti:
Facilitating the Development of IoT Applications in Smart City Platforms. IS-EUD 2019: 83-99.
Author Response
This research is very interesting and adequate to this Journal, but will be improve using a correct design of experiments as in: Application of IoT with haptics interface in the smart manufacturing industry
Answer: Thanks a lot for the comment, the article has been read and cited in the related work. That article mainly reviews the several aspects and technologies which can be used in the context of introducing IoT with haptics interface in the smart manufacturing industry. It is related to our aim and work and thus it has been taken into account as suggested by the reviewer.
And considering your case in the museum, is very important describe a danger situation as in:
Innovative Data Visualization of Collisions in a Human Stampede Occurred in a Religious Event using Multiagent Systems.
Answer: Thanks a lot for the comment, the article has been read and cited in the section in which the museum IOT App for counting people has been presented.
A multivariable analysis is important in this research.
Stefano Valtolina, Fatmeh Hachem, Barbara Rita Barricelli, Elefelious Getachew Belay, Sara Bonfitto, Marco Mesiti: Facilitating the Development of IoT Applications in Smart City Platforms. IS-EUD 2019: 83-99.
Answer: Thanks, this citation has been already included in the former version of the paper.
In general the main changes performed have been:
Strong revision of the language in all part of the paper. Strong improvement of the state of the art in introduction and related work. Introduction of a section about the development life cycle. General improvement of the method description.
Round 2
Reviewer 1 Report
The authors made most of the changes suggested by this reviewer in the previous round. The paper now has an appropriate coverage of related work and describes the development life cycle properly. However, the quality of the English remains very bad.
The paper has lots of strange sentences with broken English. Here are just a few examples from the first couple of pages:
- supporting protocols in pull for data gathering
- IOT solutions are moving forward the push protocols to event driven processes.
- a more aggregated IOT streams
- adoption of PUSH approach (no article)
- They are also called IOT Applications which have to be capable to process messages and produce reactions in real time
- For many aspects Smart City application are one of the most complex cyber physical systems, CPS, due to their complexity in terms of data, data analytics, and interfaces with the real world, physical and digital on the user interface
There are many many (tens) of other problematic sentences like these in the paper. Thus, to follow the paper, the reader must make an effort to "turn off" the part of the brain that deals with grammar and try to understand the meaning of the text even in the presence of hundreds of grammar mistakes.
The positioning of commas throughout the paper is problematic, too.
I leave to the editor the decision on whether to accept the paper in its current form (technically adequate but with hundreds of grammar problems) or to request that a real revision of the English be performed.
Minor details:
(1) Make Figure 17 larger
(2) IOT => IoT
(3) IOT Applications -> why capital A??
Author Response
The paper has been strongly improved by using the language review suggested by the MDPI editor.
The resulting paper is much more readable.
We have also verified that the problems identified by the review have been solved.
In addition:
Figure 17 has been enlarged as requested. IOT has been changed in IoT
For IoT Application we prefer to keep capital A since it recall the definition reported in Section 3.
Reviewer 2 Report
This research now is adequate by this Journal.
Author Response
Thanks for the comment.